# Structure of human Cdc45 and implications for CMG helicase function

Aline C. Simon[1], Vincenzo Sannino[2], Vincenzo Costanzo[2] & Luca Pellegrini[1]

Cell division cycle protein 45 (Cdc45) is required for DNA synthesis during genome duplication, as a component of the Cdc45-MCM-GINS (CMG) helicase. Despite its essential biological function, its biochemical role in DNA replication has remained elusive. Here we report the 2.1-Å crystal structure of human Cdc45, which confirms its evolutionary link with the bacterial RecJ nuclease and reveals several unexpected features that underpin its function in eukaryotic DNA replication. These include a long-range interaction between N- and C-terminal DHH domains, blocking access to the DNA-binding groove of its RecJ-like fold, and a helical insertion in its N-terminal DHH domain, which appears poised for replisome interactions. In combination with available electron microscopy data, we validate by mutational analysis the mechanism of Cdc45 association with the MCM ring and GINS co-activator, critical for CMG assembly. These findings provide an indispensable molecular basis to rationalize the essential role of Cdc45 in genomic duplication.

[1] Department of Biochemistry, University of Cambridge, Cambridge CB2 1GA, UK. [2] DNA Metabolism Laboratory, FIRC Institute of Molecular Oncology Foundation, 20139 Milan, Italy. Correspondence and requests for materials should be addressed to V.C. (email: vincenzo.costanzo@ifom.eu) or to L.P. (email: lp212@cam.ac.uk).

The timely and efficient duplication of our genome before cell division depends on the biochemical process of DNA replication[1]. Experimental work spanning several decades in model systems such as yeast has led to the identification of a complete list of protein factors necessary for DNA synthesis *in vitro*. However, our understanding of the complex molecular mechanisms responsible for initiation, progression and termination of DNA synthesis remains incomplete.

Cell division cycle 45 (*CDC45*) was originally identified in genetic screens for cell cycle progression mutants in yeast[2]. Its gene product was later shown to be an essential DNA replication factor[3–5]. Extensive experimental evidence has since demonstrated that cell division cycle protein 45 (Cdc45) is required throughout the process of DNA replication: it is essential for establishment of an initiation complex at DNA origins[6], and for chromosome unwinding and DNA synthesis at replication forks[7,8]. More recently, Cdc45 was shown to be an integral component of the Cdc45-MCM-GINS (CMG) assembly responsible for DNA unwinding in the replisome[9–12].

Recruitment of Cdc45 to the MCM2-7 heterohexamer is a key step in the highly regulated process that leads to the activation of helicase activity[13]. Current models of yeast CMG assembly and activation show that the MCM proteins are loaded on DNA as an inactive double hexamer in the G1 phase of the cell cycle[14]. At the beginning of S-phase, MCM phosphorylation by Dbf4-dependent kinase (DDK) promotes the recruitment of Cdc45 to the MCM ring, in a process that is mediated by the Sld3–Sld7 complex[15–17]. CMG formation is completed by recruitment of the second essential helicase co-factor, the GINS heterotetramer, in a step which requires cyclin-dependent kinase (CDK) activity, as well as the essential participation of Sld2, Dpb11 and DNA polymerase ε[18–20].

The combined effect of Cdc45 and GINS recruitment to the MCM ring, together with DDK and CDK activity, is to promote the transition from an inactive double MCM hexamer to the active CMG (Fig. 1a). The process is only known in outline, but must involve conformational remodelling of the CMG helicase to its active form, causing local melting of the DNA double-helix, steric exclusion from the MCM ring of the lagging-strand DNA template and advancement of the helicase in 3′-to-5′ direction on the leading strand. Association of replication factors MCM10 and Ctf4 with the CMG then leads to recruitment of RPA and DNA polymerase α/primase, with consequent initiation of DNA synthesis[21,22].

Structural information of the CMG assembly in apo form is available from electron microscopy studies, bound to nucleotide analogues and to a 3′-tailed double-stranded DNA[23,24], whereas atomic models from X-ray crystallography and cryoEM are available for GINS[25–27] and for MCM proteins[28–32]. These studies have revealed that co-factors Cdc45 and GINS dock onto the side of the spiral assembly of MCM subunits, straddling the dynamic interface between the adjacent AAA + ATPase domains of MCM2 and MCM5 (ref. 24).

Evidence concerning structure and function of Cdc45 remained elusive, until recent reports identified it as a remote orthologue of bacterial RecJ[33–35], a 5′-to-3′ single-stranded (ss) DNA exonuclease with multiple roles in DNA repair[36,37] and a member of the DHH phosphoesterase superfamily[38,39]. Sequence similarity with RecJ is strongest in its N-terminal DHH domain, whereas the rest of the sequence has diverged beyond reliable fold assignment. In contrast to bacterial RecJ and its putative archaeal orthologue GINS-associated nuclease, Cdc45 does not appear to have retained nuclease activity[34], rendering interpretation of its biochemical function even more uncertain.

The position of Cdc45 within the CMG, established by electron microscopy (EM) studies[23], places it opposite the juxtaposed AAA + ATPase domains of MCM2 and MCM5, a critical interface of the bipartite MCM ring that might transition through an open conformation between subsequent closed, catalytically competent states during the translocation cycle of the helicase. Thus, it has been proposed that Cdc45 is ideally placed to capture the leading-strand DNA template that might accidentally escape via the MCM2-MCM5 gate upon helicase stalling because of a replication block[24,40]. Alternative hypotheses have been formulated, suggesting that Cdc45 might act instead as a wedge, responsible for splitting strands of parental DNA during CMG progression[35].

To help rationalize the critical function of Cdc45 in eukaryotic replication, we determined the crystal structure of human Cdc45. The structure, together with accompanying functional analyses, fills a critical gap in our knowledge of the CMG complex structure and provides a necessary structural basis for our understanding of Cdc45 function in eukaryotic replication.

## Results

**Crystal structure of Cdc45**. The X-ray crystal structure of human Cdc45 was solved at 2.1 Å using the anomalous signal of crystals derivatized with the mercury compound thimerosal (Supplementary Fig. 1a and Table 1). As the amino-acid sequence of Cdc45 shows strong sequence conservation of hydrophobic residues at both N- and C-terminus, we chose to express and purify recombinant Cdc45 without affinity tags that might interfere with protein folding. Furthermore, inspection of the Cdc45 sequence showed the presence of a poorly conserved, acidic region immediately after its N-terminal RecJ-homology region. We decided to screen a series of Cdc45 constructs bearing internal deletions of different size in this low-complexity region of the protein for crystallization. A construct with an 11-residue excision (lacking amino acids S154 to I164) crystallized readily and was used for high-resolution structure determination.

**Cdc45 fold and domain structure**. *Ab initio* reconstructions of the Cdc45 shape based on small-angle X-ray scattering measurements had suggested an elongated shape with two bulging extensions on opposite sides of a central RecJ-like fold[34,35]. Crystallographic analysis reveals that human Cdc45 adopts a compact, disk-like shape, resulting from folding of the polypeptide chain into a closed arc, from its N- to the C-terminus (Fig. 1b). The only deviation from a globular fold is represented by a large helical insertion in the DHH domain, spanning amino acids L118 to S197, which projects away from the folded core of Cdc45 (Fig. 1b,c; discussed in detail later); the insertion consists of the low complexity, acidic region, which is almost entirely disordered, followed by the seven-turn helix α6 pointing away from the Cdc45 fold (Fig. 1c). The local structure of the helical insertion is maintained by hydrophobic interactions involving N- and C-terminal sequences at its base, and by DHH helix α5, that together buttress helix α6 at the point where it connects to the DHH domain.

Prediction of the tertiary structure of Cdc45 beyond its N-terminal DHH domain sequence was rendered problematic by the presence of Cdc45-specific regions. Our crystallographic analysis shows that Cdc45 bears a remarkable three-dimensional similarity to the RecJ exonuclease, which reveals their homology despite very little sequence identity (Fig. 2a and Supplementary Fig. 1b). Superposition of the atomic coordinates of human Cdc45 with the crystal structure of *Thermus thermophilus* RecJ[41,42] (PDB ID 2ZXO) allows the identification of the evolutionarily conserved RecJ/Cdc45 fold (root mean square deviation (RMSD) of 3.0 Å over 282 Cα positions; 14% sequence identity for structurally aligned residues). The fold consists of an

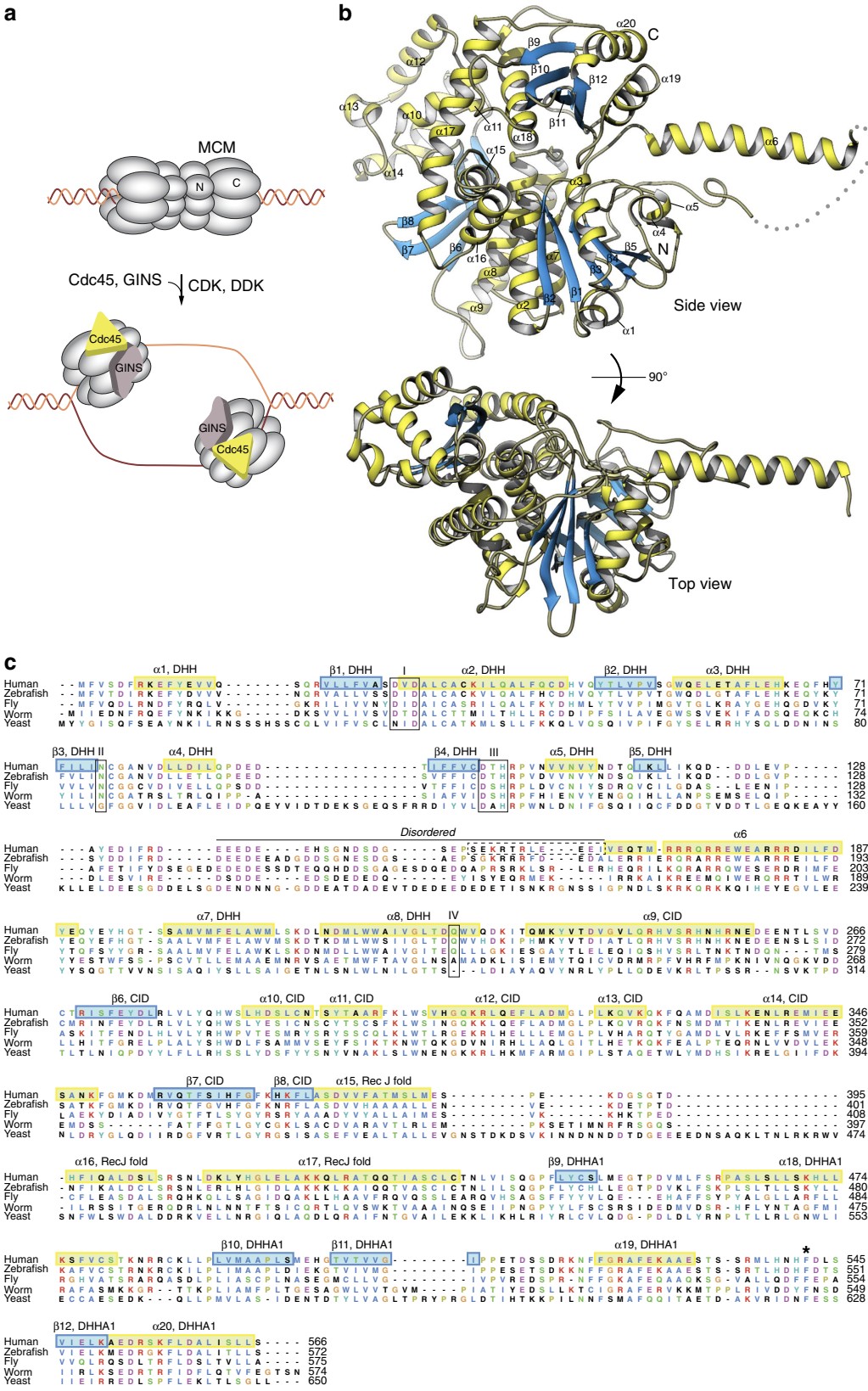

**Figure 1 | Crystal structure of human Cdc45.** (**a**) The establishment of an active DNA replication fork requires Cdc45 association with MCM2-7 and formation of the CMG helicase assembly. (**b**) Top and side views of the Cdc45 structure, drawn as a ribbon with α-helices and β-strands in yellow and light blue, respectively. The putative path of the disordered region of human Cdc45 spanning residues 137–165 is drawn as grey dots. (**c**) Multiple sequence alignment of distantly related Cdc45 proteins. The secondary structure elements are shown as yellow and light blue boxes, numbered α1-20 for α-helices and β1-11 for β-strands and annotated as belonging to the DHH or DHHA1 domains, as appropriate. The four active site motifs of the DHH domain are included in boxes and marked I–IV. The extent of the disordered region preceding helix α6 in the DHH domain is indicated. Residues that were excised from the recombinant Cdc45 protein to promote crystallization are indicated by a dashed box. The position of F542 in the DHHA1 domain is marked by an asterisk.

**Table 1 | Data collection and refinement statistics.**

| | Cdc45 |
|---|---|
| *Data collection* | |
| Space group | P3$_1$2 |
| Cell dimensions | |
| *a, b, c* (Å) | 112.8, 112.8, 143.1 |
| *α, β, γ* (°) | 90.0, 90.0, 120.00 |
| Resolution (Å) | 48.84–2.10 (2.15–2.00) |
| $R_{merge}$ | 0.105 (1.042) |
| $R_{meas}$ | 0.114 (1.131) |
| $R_{pim}$ | 0.044 (0.436) |
| $I/\sigma I$ | 11.8 (2.1) |
| CC(1/2) | 0.998 (0.649) |
| Completeness (%) | 99.8 (97.1) |
| Multiplicity | 6.6 (6.5) |
| | |
| *Refinement* | |
| Resolution (Å) | 48.84–2.10 |
| No. of reflections | 62,074 |
| $R_{work}$*/$R_{free}$† | 0.1679/0.1953 |
| | |
| *No. of non-H atoms* | |
| Protein | 5,154 |
| Ligand/ion | 39 |
| Water | 541 |
| H atoms | 4,611 |
| | |
| *B-factors* | |
| Protein | 42.70 |
| Ligand/ion | 71.10 |
| Water | 50.80 |
| | |
| *R.m.s. deviations* | |
| Bond lengths (Å) | 0.026 |
| Bond angles (°) | 0.620 |
| | |
| *MolProbity‡* | |
| Clashscore | 0.76 |
| Overall | 0.74 |

Values in parentheses are for highest-resolution shell.

$*R_{work} = \dfrac{\sum\limits_{hkl} \lvert \lVert F_{obs} \rvert - \lvert F_{calc} \rVert \rvert}{\sum\limits_{hkl} \lvert F_{obs} \rvert}$, where $F_{obs}$ and $F_{calc}$ are the observed and calculated structure-factor amplitudes, respectively.

†$R_{free}$ equals $R_{xpct}$ of the test set (5% of the data that were excluded from refinement).
‡http://molprobity.biochem.duke.edu/.

its N-terminal methionine is buried in the structure and hydrogen bonded with the main-chain carbonyl moieties of N108, N111 and I115, whereas the conserved hydrophobic tail, 1-MFVSDF-6, packs against the exposed edge of the β-sheet (Supplementary Fig. 2). These structural features indicate that N-terminal tagging is unsuitable for production of recombinant Cdc45 as it will interfere with its native fold.

In the RecJ family of DHH phosphoesterases, an additional DHHA1 domain is usually present, C-terminal to the DHH domain. The structure confirms the presence of a DHHA1-like domain in the Cdc45 sequence, spanning amino acids V439 to L566 (C-end), consisting of a four-stranded β-sheet flanked by one (α18) and two (α19 and α20) helices, respectively. Remarkably, the Cdc45 residues that coincide with the last β-strand of canonical DHHA1 domains, form a loop (amino acids H539 to L544) that mediates an intramolecular interaction with the DHH domain. The structure shows that the DHHA1 loop reaches down across the groove that separates it from the DHH domain and inserts F542 in a hydrophobic pocket on the DHH domain surface, lined by residues A24, G54, W55 and C77 and with V52, L58 as its base (Fig. 3a,b). This hydrophobic interaction is consolidated by a surrounding mesh of polar contacts, involving main-chain and side-chain hydrogen bonds of DHHA1 residue N540 with DHH residues W55 and Q56, DHHA1 F542 with DHH NW55 and N80, and DHHA1 L544 with DHH E192 (Fig. 3c). As a result of these interactions, the DHHA1 domain latches onto the DHH domain, causing the steric occlusion of the inter-domain groove (Fig. 3d), which represents the ssDNA-binding site in RecJ. The invariance of amino-acid F542 in the Cdc45 family (Fig. 1c) strongly suggests that this unexpected intramolecular interaction is a universal feature of the Ccd45 structure.

The sequence in the central region of the protein, spanning residues between the DHH and DHHA1 domains, does not belong to the RecJ fold and is unique to Cdc45. At its core lies a three-stranded antiparallel β-sheet (β6-8) sandwiched between a long helix (α9) and a helical bundle (α10-14). This domain has a prominent role in Cdc45's interactions within the CMG assembly, as it mediates all of the MCM interactions and some of the GINS interactions. Accordingly, it will be referred to in the rest of the text as CID for CMG-Interaction Domain.

**Docking of Cdc45 in the CMG structure.** During DNA synthesis, Cdc45 is present at the replication fork as an integral component of the CMG helicase complex. Our knowledge of the CMG structure derives mainly from single-particle EM reconstructions of the reconstituted holoenzyme using recombinant *Drosophila melanogaster* proteins[23,24], building on earlier mapping of MCM subunit architecture[43,44]. The EM studies have shown that the GINS and Cdc45 co-activators bind to one side of the MCM ring, creating a latch or handle structure that bridges the N-terminal collar of the MCM ring between subunits 2 and 5. In the resulting assembly, GINS and Cdc45 interact directly with each other as well as with the MCM ring. It is from this concomitant set of binary interactions that a stable Cdc45-MCM-GINS complex is formed.

The availability of crystallographic models for human GINS[25–27] has allowed its unambiguous placement in the CMG structure. Docking of GINS has shown that its MCM interactions are mediated principally by contacts of the C-terminal helical domains of Psf2 and Psf3 with the A-subdomains in the N-terminal regions of MCM 3 and 5. An additional contact is made by the N-terminal B-domain of Psf3 with the AAA+ ATPase domain of MCM3. In the CMG reconstruction, GINS occupies roughly half, or one side, of the triangular GINS-Cdc45

N-terminal DHH domain connected by a three-helix motif (α15 to α17) to a C-terminal DHH-Associated 1 (DHHA1) domain. The long α17 helix spans the distance between DHH and DHHA1 domains, whose centre of mass is separated by a distance of 30 and 32 Å in Cdc45 and RecJ, respectively (Fig. 2b,c); in RecJ, the space between DHH and DHHA1 domains is thought to represent its putative ssDNA-binding groove.

The DHH domain of Cdc45 (amino acids M1 to V229) conforms to its known architecture, with a five-stranded parallel β-sheet of 21345 topology surrounded by three α-helices on either side. The size and relative position of the three helices (α2, α7 and α8) that contribute to the active site architecture are highly conserved between Cdc45 and RecJ, whereas the helical structure on the solvent-exposed DHH side (α3, α4 and α5) shows more conformational variability. Superposition with RecJ confirms the identification of Cdc45 residues that occupy equivalent positions to active-site residues in motifs I–IV of RecJ (Fig. 1c and Supplementary Fig. 1b); the structure shows that three out of the four active-site motifs of Cdc45 contain inactivating mutations, providing a structural basis for its known lack of nuclease activity[35] (Supplementary Fig. 1c). A unique feature of Cdc45's DHH domain is that the main-chain amine of

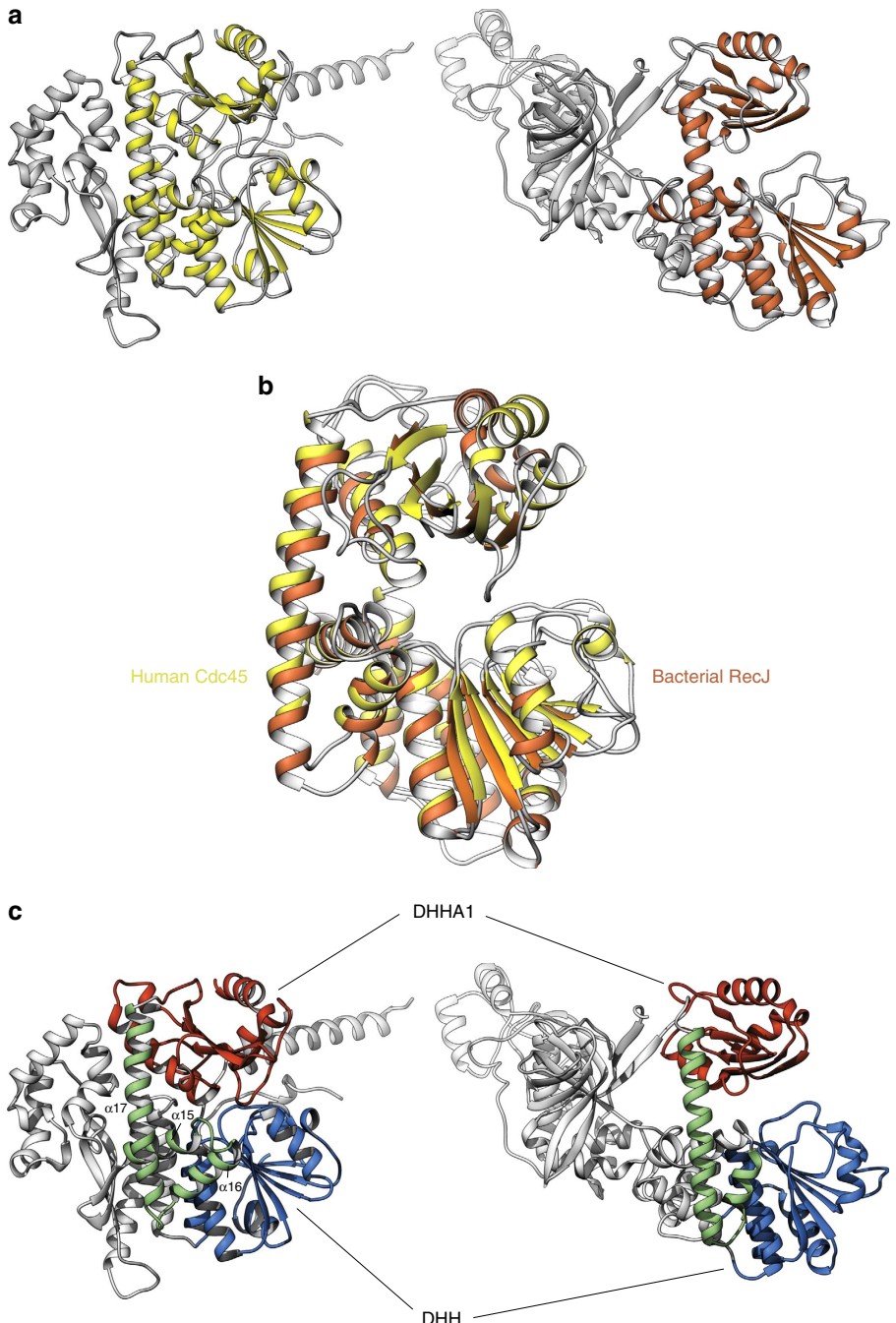

**Figure 2 | Human Cdc45 and bacterial RecJ.** (**a**) Side-by-side comparison of the crystal structures of human Cdc45 (left) and bacterial RecJ (right; PDB ID 2ZXO), drawn as ribbons. The RecJ-homology fold is coloured yellow and orange in Cdc45 and RecJ, respectively, whereas the rest of the structure is coloured light grey. (**b**) Superposition of the RecJ-homology folds of Cdc45 and RecJ, coloured as in **a**. (**c**) Side-by-side comparison of Cdc45 and RecJ, highlighting the position of their DHH and DHHA1 domains, relative to the rest of their structures. The DHH and DHHA1 domains are coloured blue and red, respectively; helices 15 to 17, which connect the DHH and DHHA1 domains in human Cdc45 are coloured in light green.

shape protruding from the MCM ring, leading to the indirect identification of the position occupied by Cdc45 in the helicase assembly.

We docked the crystal structure of human Cdc45 into the cryoEM reconstruction at 7.4 Å of fly CMG bound to DNA and ATPγS (EMDB ID 3318)[45]. Some of the outstanding features of the Cdc45 structure are clearly identifiable in the cryoEM map, such as the DHH and DHH1 domains, and the protrusion represented by helix α6, allowing for the unambiguous placement of the crystal structure in the map, using the 'Fit to Segments'

option in UCSF Chimera[46]. The result of the docking shows that the narrow edge of Cdc45's disk-like structure spanning the DHH and CID domains forms a continuous interface with GINS and the N-terminal collar of the MCM ring, respectively (Fig. 4). This mode of Cdc45 interaction with the CMG leaves both faces of the Cdc45 disk-like shape exposed to solvent and poised in principle for further interactions.

Relative to the sixfold axis of the MCM heterohexamer in the CMG complex, Cdc45 is tilted by ~40°. This mode of Cdc45 association imparts volume and accessibility to the outer chamber

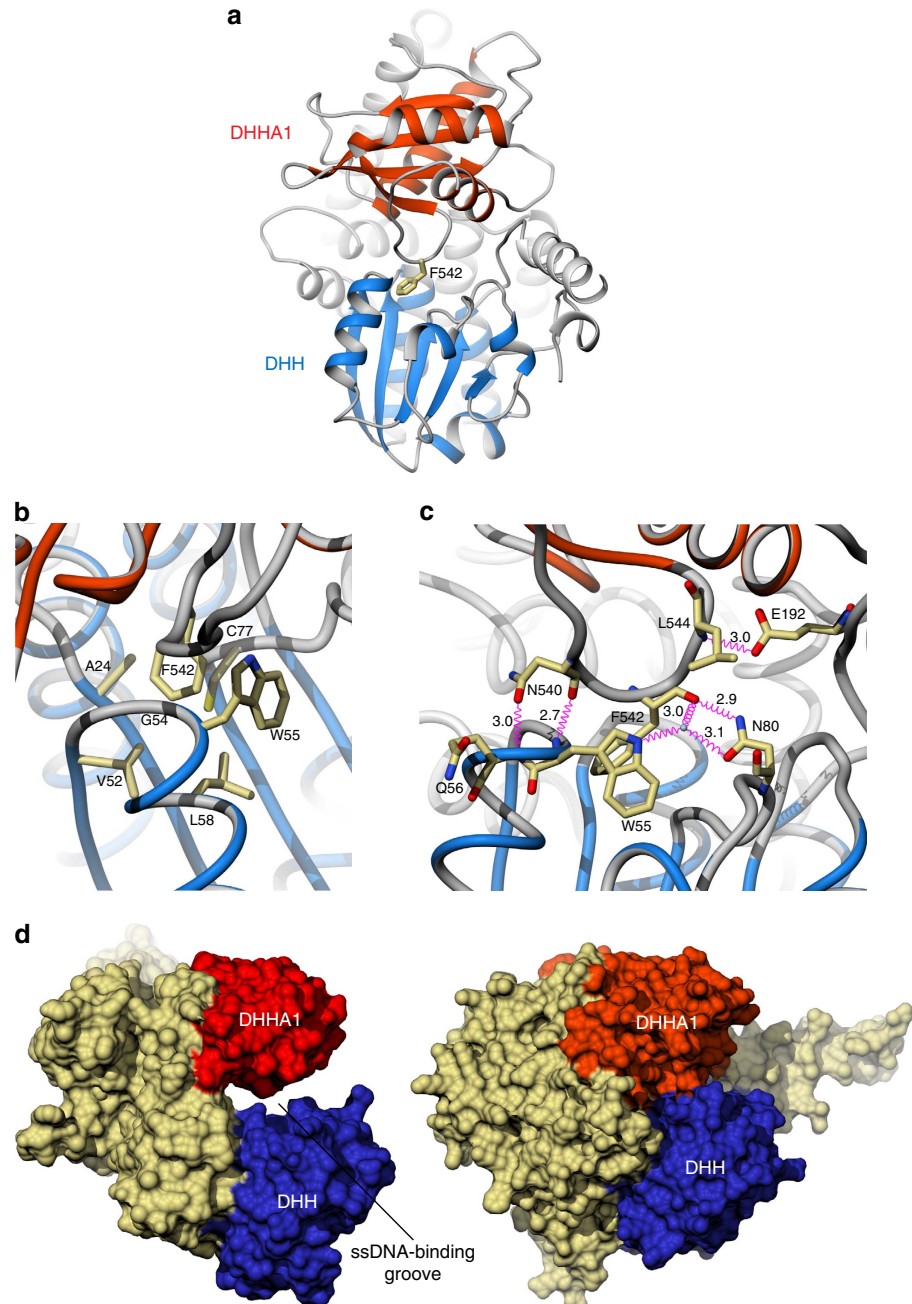

**Figure 3 | Intramolecular association of the DHH and DHHA1 domains.** (**a**) Edge-on view of Cdc45, showing the position of the DHHA1 loop and the sidechain of F542, which mediates the intramolecular association of the DHHA1 and DHH domains in Cdc45. The DHH and DHHA1 domains are coloured light blue and red, respectively. Panels **b** and **c** show details of the hydrophobic and hydrophilic interactions at the DHH–DHHA1 interface, respectively. Hydrogen bonds are drawn in pink, and solvent molecules as small grey spheres. The distance of the polar contacts in panel **c** are shown in Ångström. (**d**) Side-by-side comparison of the RecJ (left; PDB ID 2ZXO) and Cdc45 (right). The DNA-binding groove of RecJ becomes inaccessible in Cdc45 because of the intramolecular association of its DHH and DHHA1 domains. The solvent-accessible surface of RecJ and Cdc45 is shown, coloured in light brown, whereas the DHH and DHHA1 domains are coloured blue and red, respectively.

formed by the GINS–Cdc45 sub-complex in the closed, planar form of the CMG. Docking Cdc45 in the CMG reconstruction further shows that the DHHA1 domain and inter-domain helices α15-17 of Cdc45's RecJ-like fold face the C-terminal ring of MCM ATPase domains, directly opposite to the interface or gate between MCM subunits 2 and 5.

**The Cdc45–MCM interface.** In the CMG, Cdc45 binds to the N-terminal tier of the MCM ring, engaging the A-subdomains of both MCM 2 and 5. A large portion of Cdc45's CID is dedicated to the interaction with the MCM ring (Fig. 5a): the α10-14 bundle in particular appears to have a paramount role in the interaction, as it is wedged between MCM2 and 5, while the exposed edge of the β6-8 sheet extends the interaction surface with specific contacts to MCM5 (Fig. 5b).

To analyse the Cdc45–MCM interactions in detail, we built homology models of the A-subdomains for human MCM2 and MCM5 using the Phyre2 server[47]. The models were then superimposed on the same domains of a single hexamer of the

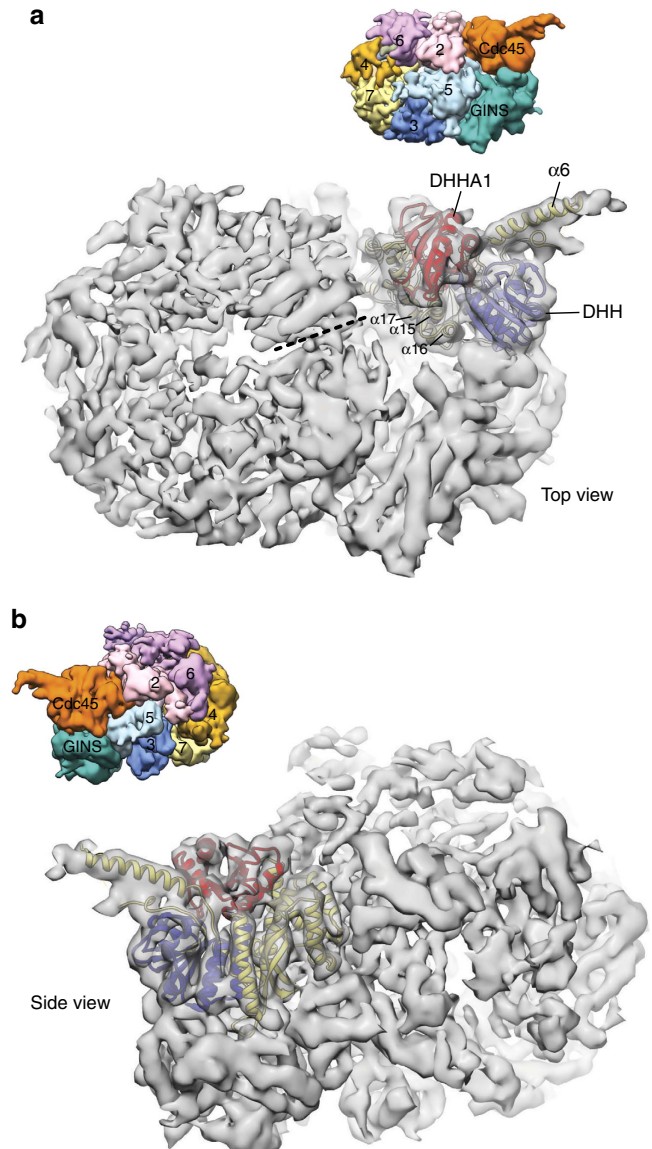

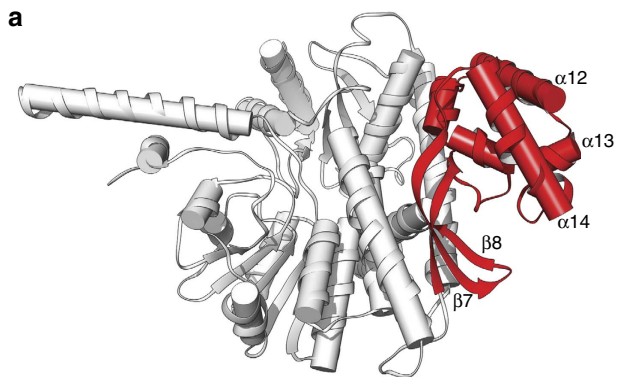

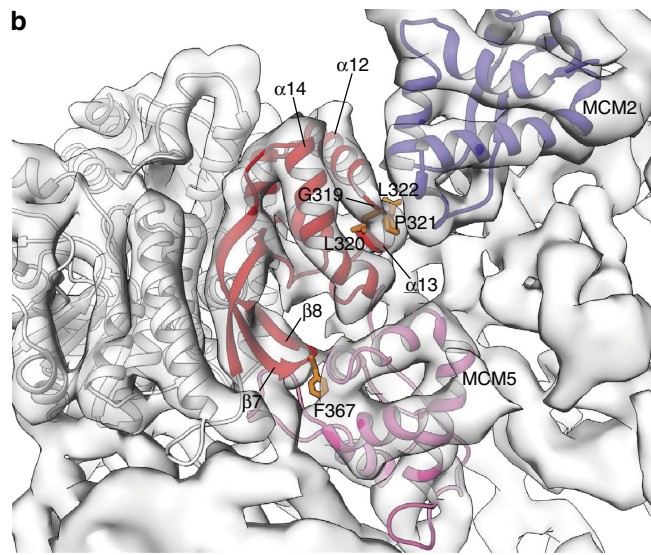

**Figure 4 | Docking of Cdc45 in the cryoEM map of the CMG.** Panels **a** and **b** show top and bottom views of the Cdc45 structure fitted into the cryoEM map of fly CMG bound to DNA and ATPγS at 7.4 Å (EMDB ID 3318). Cdc45 is drawn as a ribbon, with the DHH and DHHA1 domains coloured blue and red, respectively, and the CMG map is shown as a grey, transparent envelope. The inset in each panel shows the CMG map, segmented and coloured to illustrate the position of each of its MCM2-7, GINS and Cdc45 subunits.

**Figure 5 | The Cdc45–MCM interface.** (**a**) The MCM-binding domain of Cdc45 is highlighted in red, relative to the rest of the Cdc45 structure in light grey. The structure is drawn as ribbons, with cylinders for α-helices. (**b**) The interface of Cdc45 with MCM2-7. The Cdc45 structure and homology models for the A-subdomain of MCM2 and MCM5 are shown as ribbons, docked in the map of fly CMG bound to DNA and ATPγS at 7.4 Å (EMDB ID 3318). The MCM-binding domain of Cdc45 is shown in red, the A-subdomains of MCM2 and MCM5 are shown in purple and pink, respectively, and the CMG map is drawn as a transparent envelope in light grey. The side chains of residues 319-GLPL-322 and F367, which are at the interface with MCM2 and MCM5, respectively, are also shown.

yeast MCM2-7 structure (PDB ID 3JA8)[32] that had been docked into the CMG density, and their orientation was optimized by further fitting.

Inspection of the resulting Cdc45–MCM interfaces highlights two prominent points of contact (Fig. 5b). The first involves the hydrophobic motif 319-GLPL-322 that links helices α12 and α13 of Cdc45's MCM-binding region. The motif provides a point of close contact with the exposed vertex in the helical triangle of the A-subdomain of MCM2. In the temperature-sensitive *cdc45-1* allele that led to Cdc45 identification in yeast, the invariant glycine residues 367, equivalent to G319 of human Cdc45, was mutated to Asp (G367D)[4]. Our structural analysis shows that the biochemical cause for the genetic defect is likely to be a defective interaction of Cdc45 with MCM2.

The second main interface between Cdc45 and MCM involves the β7-8 hairpin and the MCM5 A-subdomain. Docking shows that conserved F367, which occupies an unusual, solvent-exposed position at the tip of the hairpin, can become buried in a pocket of aromatic side chains formed by Y42, F51 and F80 of MCM5 (Supplementary Fig. 3). Interestingly, the electron density map of the β7-β8 hairpin provides an indication that it can occupy multiple positions; such conformational mobility might play a functional role in establishing the correct Cdc45–MCM5 interaction.

Structural analysis of archaeal MCMs in ring and filament assemblies has provided evidence that the relative orientation of the A-subdomain can change[28,31,48]. The best-fitting position of the A-subdomain in the CMG map, and the one compatible with Cdc45 binding, corresponds to that observed in the structures of the N-terminal hexameric SsoMCM[28] and yeast MCM2-7 (ref. 32). The *mcm5-bob1* P83L mutation of yeast MCM5, which bypasses the requirement for MCM phosphorylation by DDK[49], presumably facilitates Cdc45 loading on chromatin by

stabilizing this conformation of the A-subdomain in eukaryotic MCM5.

**The Cdc45—GINS interface**. The Cdc45 and GINS co-activators contact each other directly in the CMG assembly[23]. Biochemical reconstitution experiments have provided evidence that the Cdc45–GINS interaction is important for their incorporation into the CMG[50]. Docking of Cdc45 in the CMG map reveals the structural elements that mediate its interaction with GINS. The GINS-binding surface of Cdc45 is composite: the main part is provided by the DHH domain augmented by α16 of the RecJ fold, which contacts the B-domain of GINS subunit Psf1, whereas a separate epitope formed by an acidic loop linking CID elements α9 and β6 interacts with Psf2 (Fig. 6a).

The C-terminal B-domain of GINS subunit Psf1 is essential for chromatin binding and DNA replication activity[27,51]. In the crystal structure of human GINS, the Psf1 B-domain is disordered, as a consequence of its flexible tethering to the GINS core structure[25]. However, the EM map of the CMG shows clearly a density volume for the Psf1 B-domain, which positions it at the interface with Cdc45, in agreement with biochemical evidence that the domain mediates a contact with Cdc45 that is essential for CMG assembly[50].

Although the resolution of the CMG map is not sufficient to determine unambiguously the orientation of the Psf1 B-domain, it is clear that it makes multiple contacts with secondary-structure elements on the side of Cdc45's DHH domain (Fig. 6b). The exposed β2 strand at the edge of the DHH domain appears to be at the centre of the interface, where it would be favourably positioned to associate with the beta hairpin of the Psf1's B-domain. The Psf1–Cdc45 interface is completed by DHH helices α2 and α3 that flank β2 and by helix α16 of the RecJ fold. A second, separate interface is provided by an extended, acidic loop linking α9 and β16 of Cdc45, which docks against the Psf2 subunit of GINS, at the point of junction between its N-terminal B-domain and its helical domain.

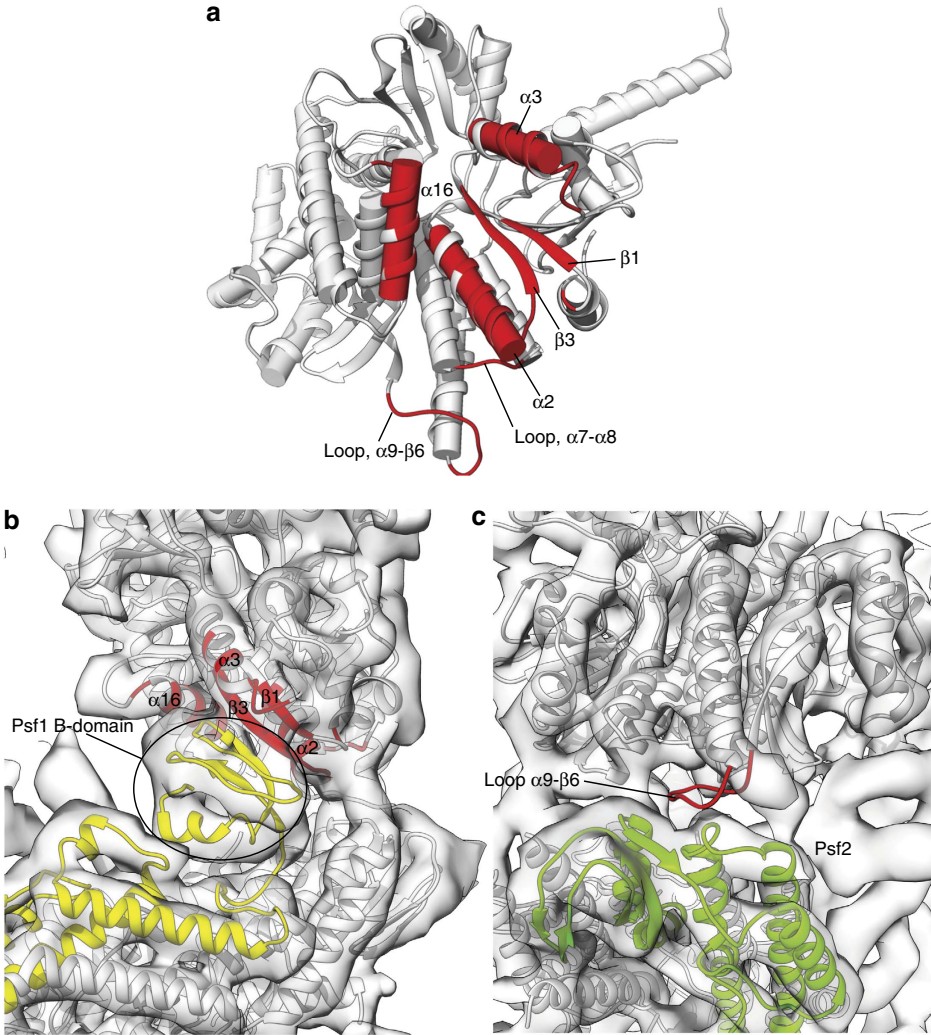

**Figure 6 | The Cdc45–GINS interface.** (**a**) The structural elements of the Cdc45 structure that come into contact with GINS in the CMG complex are highlighted in red, relative to the rest of the Cdc45 structure in light grey. The structure is drawn as ribbons with cylinders for α-helices. (**b**) View of the Cdc45–GINS interface that focuses on the interaction of the Psf1 B-domain with the DHH domain. Cdc45 is coloured as in **a**. The Psf1 subunit of GINS is shown in yellow, whereas the rest of the GINS structure (PDB ID 2Q9Q) is drawn in light grey. The Psf1 B-domain was modelled in Phyre2 and docked manually in the map of fly CMG bound to DNA and ATPγS at 7.4 Å (EMDB ID 3318); given the limited resolution of the map, its orientation relative to Cdc45 must be considered tentative. (**c**) View of the Cdc45–GINS interface that emphasises the contact between Psf2 and the acidic loop 256-NEDEENTLSVDC-267, linking α9 and β6 of the CID. Cdc45 is coloured as in **a**. The Psf2 subunit of GINS is shown in green, whereas the rest of the GINS structure is drawn in light grey.

**Structure—function analysis of Cdc45**. We set out to probe some of the most interesting features of the Cdc45 structure, by testing the ability of site-specific Cdc45 mutants to support efficient DNA synthesis during DNA replication in *Xenopus* egg extracts.

Our structural analysis had identified areas on the Cdc45 surface that mediate the interaction with MCM2-7 and GINS within the CMG complex. We designed and prepared

recombinant versions of several *Xenopus* Cdc45 proteins, containing mutations aimed at disrupting Cdc45's association with MCMs and GINS (Supplementary Fig. 4a,b). For the interaction with MCM, we targeted for mutagenesis the conserved 319-GLPL-322, I334 and 367-FK-368 motifs that were shown by docking to be at the interface with MCM2 and MCM5, respectively (Cdc45 mutants 'MCM1-3'; Fig. 7a). For the

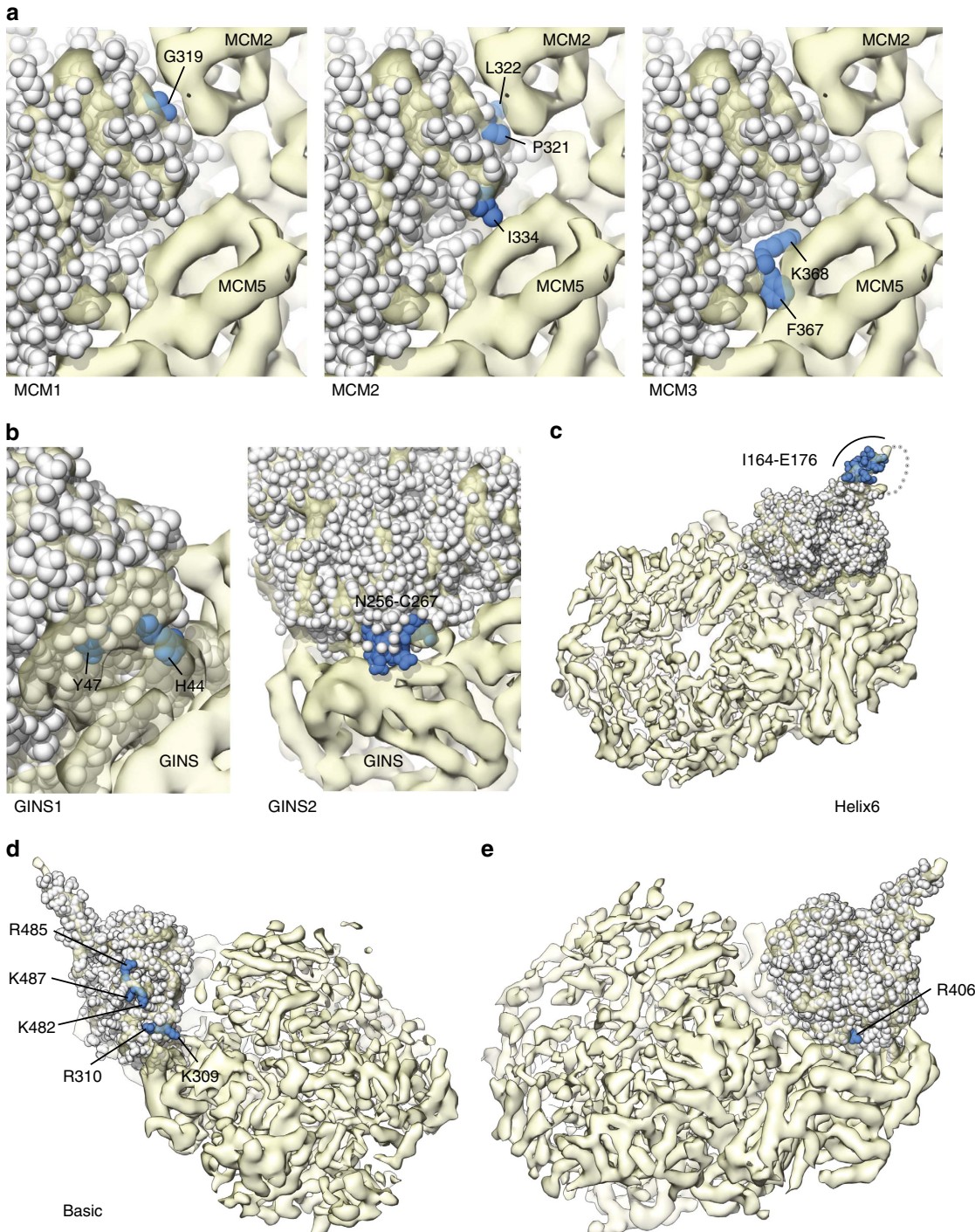

**Figure 7 | Structure-guided mutagenesis of Cdc45.** The Cdc45 mutants chosen for functional analysis in *Xenopus* egg extracts are shown in the crystal structure of human Cdc45. The panels illustrate the position of the residues targeted for mutation, shown in blue on white space-fill model of the Cdc45 structure, docked in the cryoEM map of the of fly CMG bound to DNA and ATPγS at 7.4 Å (EMDB ID 3318), coloured in light brown. (**a**) Cdc45 mutants MCM1 to 3. (**b**) Cdc45 mutants GINS1 and GINS2. (**c**) Cdc45 mutant Helix6. (**d**) Cdc45 mutant Basic. (**e**) Cdc45 mutant targeting R406 (R407 in *Xenopus* Cdc45). The portion of the CMG map relative to the Psf1 B-domain was omitted for clarity.

interaction with GINS, we chose to target DHH residues H44 and Y47, facing the B-domain of GINS subunit Psf1, and the acidic loop between α9 and β6 that is in contact with Psf2 (Cdc45 mutants 'GINS1-2'; Fig. 7b). Mutagenesis altered the chemical nature of the amino acid, as in MCM mutants 1 to 3 and GINS1, or affected the wholesale replacement of the targeted region, as in the GINS2 mutant. In addition, we targeted some of the noticeable structural features of Cdc45 for mutagenesis. Thus, Cdc45 mutant 'Helix 6' truncated part of the exposed helix α6 pointing away from the CMG (Fig. 7c), Cdc45 mutant 'Basic' reversed the charge of a patch of solvent-exposed basic residues in the CID (Fig. 7d). We also chose to mutate to alanine R406 in human Cdc45 (Fig. 7e) (R407 in *Xenopus* Cdc45) that had been implicated in DNA binding in the fly CMG[40]. The crystal structure of human Cdc45 shows that R406 would be in a suitable position to interact with a DNA strand traversing the Cdc45-GINS channel of the CMG. The complete list of all *Xenopus* Cdc45 mutants, describing affected amino acids, type of alteration and equivalent residues in the human protein is reported in Table 2.

The ability of the Cdc45 mutants to support DNA replication was assessed in a DNA synthesis assay, supplementing the egg extracts with purified recombinant protein at a concentration that out-competed the chromatin association of endogenous Cdc45. The concentration of recombinant wild-type Cdc45 that displaced the endogenous protein was in agreement with an earlier study[52] (Supplementary Fig. 4c). Replacement of endogenous Cdc45 with recombinant wild-type protein caused higher levels of DNA synthesis relative to the endogenous protein (Supplementary Fig. 4d), an observation that is compatible with the increased chromatin recruitment of Pol α reported in the same study. Both sets of Cdc45 mutants targeting the GINS and MCM interfaces showed strong inhibitory effects on DNA synthesis, compared with the level of DNA synthesis observed with the wild-type recombinant protein (Fig. 8a). The Cdc45 mutants that do not target its CMG interfaces showed less prominent but nevertheless clearly noticeable effects in reducing levels of DNA synthesis, with the most pronounced defect observed for the 'Basic' mutant (Fig. 8a).

To gain further insight into the mechanism by which the structure-based mutations affect DNA replication, we analysed the chromatin-bound fractions of the egg extracts at different time points during replication in the presence of each Cdc45 mutant. This chromatin recruitment assay determines the ability of the recombinant proteins to replace the endogenous Cdc45 on chromatin and it therefore represents a *bona fide* measurement of their incorporation into the CMG. Furthermore, the assay provides information on the kinetics of replication for each Cdc45 mutant, by providing information on the timing of recruitment of essential replication factors such as Pol α and the GINS subunit Psf3.

All mutants targeting the CMG interfaces of Cdc45 showed various levels of reduction in Pol α and GINS binding to chromatin, ranging from severe (GINS1) to moderate (MCM3), as shown in the chromatin recruitment assay of Fig. 8, in agreement with the results of the DNA replication assays. Furthermore, the chromatin recruitment assay highlighted an intriguing difference in the ability of Cdc45's MCM and GINS mutants to displace endogenous Cdc45: whereas Cdc45 mutants GINS1-2 were loaded efficiently onto chromatin, Cdc45 mutants MCM1-3 were not stably loaded or they dissociated rapidly. The remaining Cdc45 mutants Helix 6, Basic and R407A showed altered recruitment levels for Pol α and GINS, without a clear overall trend, with a marked increase for Helix6, a slight decrease for Basic and no appreciable differences for the R407A mutant (Fig. 8). Interestingly, whereas Helix6 and R407A were able to replace partially the endogenous Cdc45, Basic had a pronounced defect in chromatin loading, comparable to that of MCM2 and MCM3 (Fig. 8).

The accompanying kinetic analysis (Supplementary Fig. 5) showed a tendency of the interface mutants to cause delayed recruitment of Pol α to chromatin. Relative to what observed for endogenous and wild-type Cdc45, where Pol α's recruitment peaked at 30', the interface mutants exhibited a noticeable lag, which was most conspicuous in the case of GINS1 and MCM1-2, where Pol α peaked at 60', and less pronounced for GINS2 and MCM3. In contrast, the Cdc45 mutants Helix6, Basic and R407A showed normal kinetics of Pol α and GINS recruitment to chromatin.

## Discussion

Our understanding of the physiological role of Cdc45 has remained incomplete since its discovery as an essential gene in DNA replication[2], partly due to lack of structural information. The crystallographic model of human Cdc45 presented here, together with its functional analysis, contributes an essential building block to our growing knowledge of the architecture of the eukaryotic replisome.

The availability of a high-resolution atomic model for Cdc45 allows the rationalization of some of the genetic evidence concerning its role in DNA replication that has accumulated over the years from studies on fungal model systems. Thus, our work shows that the biochemical basis for the replication defect of the yeast *Cdc45-1* strain bearing the G367D mutation (G319 in human Cdc45) is likely to be a weakened association with the MCM2-7N-terminal ring (Fig. 5b). Conversely, the synthetic lethality of *Cdc45-10* (G510D, I430 in human Cdc45) with a camptothecin-mimetic strain[53] is probably caused by local destabilization of the RecJ fold. Interestingly, the weakened affinity towards Sld3 caused by the L131P (V106 in human Cdc45) mutation of the *Cdc45-27* allele[54,55] points to an

**Table 2 | *Xenopus* Cdc45 mutants tested in the DNA replication assay.**

| Name | Mutation | | Target |
|------|----------|--|--------|
| MCM1 | G320N | (G319) | Cdc45-MCM2 |
| MCM2 | P322E, L323E, I335E | (P321, L322, I334) | Cdc45-MCM2 |
| MCM3 | F368E, K369E | (F367, K368) | Cdc45-MCM5 |
| GINS1 | H44E, Y47A | | Cdc45-Psf1 |
| GINS2 | N257-C268 -> GGSGGS | (N256–C267) | Cdc45-Psf2 |
| Basic | K310E, K311E, K483E, R486E, K488E | (K309, R310, K482, R485, K487) | CID's basic patch |
| Helix6 | A164-E177 -> GSG | (I164–E167) | DHH helix 6 |
| R407A | R407A | (R406) | DNA binding |

For each mutation, the number of the corresponding residues for the human Cdc45 is reported in brackets, if different from the *Xenopus* protein. In the GINS2 construct, amino acids N257 to C268 of *Xenopus* Cdc45 were replaced with the sequence GGSGGS. In the Helix6 construct, amino acids A164 to E177 were replaced with the sequence GSG.

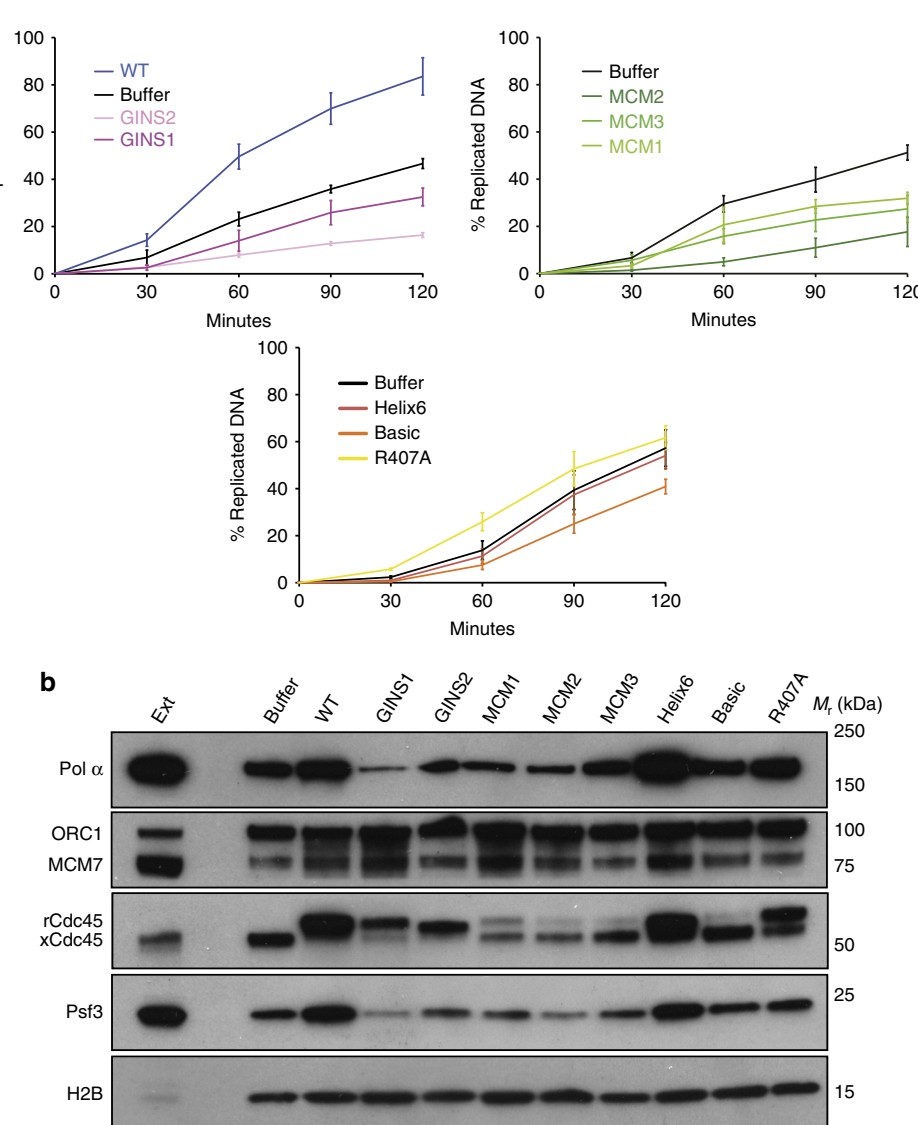

**Figure 8 | Structure-based functional characterization of Cdc45.** (**a**) Wild-type (WT) and mutant *Xenopus* Cdc45 proteins were tested for their effect on DNA synthesis during DNA replication in *Xenopus* egg extracts. Each panel shows the results of the DNA synthesis assay, given as percentage over time of replicated DNA relative to the input (6,000 nuclei per µl). The assay for each mutant was performed in triplicate. The error bars represent the standard errors. The result of the assay for buffer addition is also shown. (**b**) Chromatin association of WT and mutant *Xenopus* Cdc45 proteins during DNA replication and its effect on the chromatin binding of other replication factors. Immunoblotting was carried out on isolated chromatin fractions after incubation for 30 min of sperm nuclei (4,000 nuclei per µl) in *Xenopus* interphase extracts supplemented either with buffer, WT or mutant Cdc45 protein at a final concentration of 100 ng µl$^{-1}$. 1 µl of interphase extracts were loaded as input lane (Ext). Polyclonal rabbit anti-*Xenopus* Cdc45 was used to recognize the endogenous Cdc45 (xCdc45), as well as the recombinant WT and mutant proteins (rCdc45). Specific antibodies (described in the Methods) were used to evaluate the chromatin association level for Pol α, ORC1, MCM7, Psf3 and H2B.

involvement of DHH's helical subdomain containing helix α6 in the Cdc45–Sld3 interaction (Supplementary Fig. 6). Other phenotypes, such as the CDK-independent replication promoted by the Cdc45 allele *JET1* cannot be easily rationalized by the structure, as the causative H22Y mutation is located in an exposed loop of the DHH domain that is absent in metazoan Cdc45 sequences (Fig. 1c).

In its association with the bipartite MCM ring structure, Cdc45 wedges its CID between the adjacent A-subdomains of MCM5 and MCM2. In this mode of binding, Cdc45 fastens to and further stabilizes the N-terminal ring of the MCM2-7 hetero-hexamer, while leaving the C-terminal ring of AAA+ ATPase domains free to move in concert as required for helicase activity[24]. The limited size of the binary interfaces that keep

Cdc45 associated with the rest of the CMG supports the concept that an intermediate 'placeholder' factor is required during CMG assembly to maintain stability of a proto-CMG containing only MCM2-7 and Cdc45. In yeast, such a role as Cdc45 chaperone has been ascribed to Sld3 (refs 55,56) and might be shared with its metazoan orthologue Treslin[57,58].

The behaviour of the interface mutants in the chromatin recruitment assay is in agreement with the current paradigm of a stepwise mechanism of CMG assembly, whereby Cdc45 is recruited first to the MCM ring, followed by incorporation of GINS. Consequently, a defect in the interaction with MCM2-7 would negatively affect the stable loading of Cdc45 onto chromatin, whereas a defect in its ability to associate with GINS would not or only to a lesser degree, which is the observed

behaviour of the MCM and GINS mutants, respectively. Although both sets of interface mutants are designed to interfere with CMG assembly, the underlying mechanism is likely to be different. For the GINS mutants, MCM binding would prevent subsequent recruitment of GINS and completion of CMG assembly. We surmise that, for the MCM mutants, CMG assembly would be impaired by competitive inhibition of endogenous Cdc45, resulting from normal transport of the mutant proteins to chromatin coupled to their inability to associate stably with the MCM ring. Of the remaining Cdc45 mutants Helix 6, Basic and R407A, the pronounced defect in chromatin loading of Basic suggests that this mutant might be unable to engage efficiently with the chaperone machinery responsible for loading of Cdc45 onto chromatin. A complete understanding of the different functional effects observed for the non-interface mutants Helix6, Basic and R407A will require further investigation of Cdc45's role in CMG function and mechanism of assembly.

A remarkable feature of Cdc45's association with MCM and GINS, highlighted by docking the crystal structure in the EM map of the CMG, is the large extent of its surface area, comprising both faces of the disk-like Cdc45 shape as well as much of its outer rim, are exposed to solvent and poised for further interactions. A striking example of this is the helical protrusion comprising the long helix α6, which projects away from the Cdc45 and the rest of the CMG. The mode of Cdc45 association within the CMG is therefore compatible with Cdc45's reported involvement in several additional interactions with replisome factors.

The specific association of GINS and Cdc45 with the MCM ring is required for helicase activity and replisome assembly. Conversely, partial CMG disassembly by controlled dissociation of GINS or Cdc45 has been proposed as a way of transiently slowing or pausing the helicase during replicative stress[59,60]. These models postulate that Cdc45 might act as molecular brake, by holding on to the unwound DNA behind the fork and preventing excessive generation of ssDNA, while mediating rapid CMG reassembly once the replication roadblock has been cleared[59,60].

Molecular details of a possible association of Cdc45 with fork DNA during normal replication or under replicative stress are currently missing, as the exact path traced by the unwound strands of parental DNA through the CMG and the rest of the eukaryotic replisome remains to be established. Indirect evidence of a role for Cdc45 in interacting with fork DNA comes from its position in the CMG, poised to capture any DNA that might escape through the MCM2-MCM5 gate, as well as from its structural similarity to the RecJ exonuclease. In addition, biochemical evidence of DNA binding by isolated Cdc45 has been obtained[34,35,59] and we have confirmed that human Cdc45 possesses a clear, but weak and unspecific affinity for a variety of DNA substrates (Supplementary Fig. 7).

The structural similarity of Cdc45 with RecJ uncovered by our work would naturally suggest a similar mode of interaction with DNA, whereby ssDNA is threaded between DHH and DHHA1 domains. Surprisingly, our data show that this DNA-binding mode is precluded to Cdc45, as its putative DNA-binding groove is made sterically inaccessible by the intramolecular association of its N-terminal DHH and C-terminal DHHA1 domains. The structural basis for this unexpected conformation is remarkable and appears highly specific, as its main determinant is represented by the insertion of invariant F542 located in an exposed loop of the DHHA1 domain into an hydrophobic pocket of the DHH domain, pinning the two domains together. It is possible that the Cdc45 conformation observed in the crystal structure might represent a closed state, requiring specific conditions to convert into an open form compatible with DNA

binding. Although this remains an exciting possibility, the large size of the intramolecular DHH–DHHA1 interface indicates that such conformational change would likely require extensive compensatory interactions. The excellent fit of the Cdc45 crystal structure in the CMG map indicates that such a 'closed state' would represent the default conformation of Cdc45 in the CMG assembly. Alternatively, the intramolecular association between DHH and DHHA1 domains might represent a constitutive conformation, required to satisfy a structural requirement for extra stability that might be important to Cdc45's role as architectural component of the replisome. While losing its ancestral mode of DNA binding, Cdc45 might have evolved an alternative DNA-binding surface, which remains to be determined.

## Methods

**DNA constructs for human Cdc45.** Full-length human Cdc45 was PCR amplified from IMAGE clone 2964592 (Source BioScience) and cloned into a bacterial pRSFDuet-1 expression plasmid (Novagen) via unique NdeI and XhoI sites. For crystallization, an untagged Cdc45 construct lacking residues 154–164 was generated using overlapping extension PCR and cloned into a pRSFDuet-1 expression plasmid (Novagen) via unique NdeI and XhoI sites. For biochemical studies, a TEV protease site, followed by a His$_{10}$-affinity tag and a 1x Flag tag was introduced using PCR primer extension at the end of the Cdc45 open reading frame.

**DNA constructs for Xenopus Cdc45.** Full-length Xenopus laevis Cdc45 was PCR amplified from a chemically synthesized GeneArt DNA template that had been codon optimized for expression in E. coli (Fisher Thermo Scientific) and cloned into a bacterial pRSFDuet-1 T7 expression plasmid (Novagen) fused to a C-terminal, TEV-cleavable His$_{10}$-Flag tag as described for full-length human Cdc45. For single and multiple point mutations, G320N (MCM1); H44E, Y47A (GINS1); F368E, K369E (MCM2), P322E, L323E, I335E (MCM3), K310E, K311E, K483E, R486E, K488E (basic) and R407A were introduced by rounds of Quikchange mutagenesis (Thermofisher) into the wild-type, full-length construct. Two deletion mutant constructs were prepared by overlapping extension PCR: in the first construct (GINS2), residues E258-D267 were replaced with GGSGGS, whereas in the second construct (helix 6), residues 165–176 were replaced with GSG.

**Purification of human Cdc45 for X-ray crystallography.** Untagged human Cdc45 carrying an internal deletion (residues 154–164) was overexpressed in E. coli strain BL21(DE3)Rosetta2 with isopropyl-β-D-thiogalactoside induction and overnight expression at 20 °C in Turbo broth. After overexpression, 4–6 l of cells were harvested and resuspended in 20 mM HEPES, pH 7.5, 160 mM NaCl, 5% (w/v) glycerol and protease inhibitors (Sigma). Cells were lysed via sonication and the crude extract was clarified by centrifugation.

The supernatant was applied to a 5-ml HiTrap Q HP anion exchange column (GE Healthcare). The column with bound Cdc45 was washed until the UV$_{280}$ absorption reached baseline levels and the protein was eluted with a buffer gradient of 0.16–0.5 M NaCl over 30 CVs. Fractions with eluted Cdc45 were pooled, the salt concentration was adjusted to 100 mM NaCl, and the sample was cleared by centrifugation at 16,000$g$ for 10 min at 4 °C.

The protein supernatant was bound to a 5-ml Heparin HiTrap HP column (GE Healthcare) equilibrated in 10 mM HEPES, pH 7.2, 100 mM NaCl and 10% (w/v) glycerol, and the protein was eluted with a stepwise gradient against 20 mM HEPES, pH 8.0, 1 M NaCl.

Peak fractions collected at around 13% elution buffer were pooled, adjusted to a conductivity below 160 mM NaCl and loaded onto a 1-ml RESOURCE Q anion exchange column equilibrated in 20 mM HEPES, pH 8.0, and 160 mM NaCl. Cdc45 was eluted in a stepwise gradient against 20 mM HEPES, pH 8.0, and 1 M NaCl at around 10% elution buffer.

Fractions containing pure Cdc45 were loaded on a Superdex S75 16/60 column (GE Healthcare) equilibrated in 20 mM HEPES, pH 7.2, 150 mM NaCl and 5% (w/v) glycerol and peak fractions were pooled, concentrated to 11.5 mg ml$^{-1}$, flash frozen in liquid nitrogen and stored in small aliquots at −80 °C.

**Purification of His-Flag-tagged Cdc45 proteins.** All His-Flag-tagged Cdc45 constructs were expressed as described for the untagged Cdc45 construct. Cells were resuspended in buffer containing 20 mM HEPES, pH 7.0, 100 mM NaCl, 10 mM imidazole and 1 mM dithiothreitol (DTT) in the presence of Benzonase (Novagen, 3 U ml$^{-1}$ of resuspended pellet) and protease inhibitors (Sigma) and lysed via sonication. Subsequently, the salt concentration was adjusted to 500 mM NaCl and the crude extract was clarified by centrifugation.

The supernatant was applied to a 4-ml column of nickel agarose resin (Sigma) using gravity flow and the column with bound human or *Xenopus* Cdc45 was washed in buffer supplemented with 40 mM imidazole. Cdc45 elution was performed with buffer supplemented with 60–200 mM imidazole. The human Cdc45 protein was incubated with TEV protease overnight under gentle rotation at 4 °C, for tag cleavage. For wild-type and mutant *Xenopus* Cdc45 proteins, the TEV cleavage step was omitted.

After the initial step of Nickel-affinity chromatography, Cdc45 proteins were further purified by heparin, ion-exchange and gel-filtration chromatography, as described for the untagged Cdc45 construct.

**Crystallization and structure determination.** Cdc45 crystals were grown by vapour diffusion in sitting drop, mixing Cdc45 protein at 11.5 mg ml$^{-1}$ with an equal volume of buffer condition C2 of the Morpheus HT 96 crystallization screen (Molecular Dimensions) at 19 °C.

Cdc45 crystals appeared within 1–2 days and grew to full size over the course of 1 week. For structure determination, native crystals were transferred into fresh crystallization buffer without imidazole and were soaked for 16 h with thimerosal at 0.5 mM (final drop concentration). Subsequently, crystals were back-soaked for 2 min in fresh crystallization buffer and flash-frozen in liquid nitrogen.

X-ray diffraction data for derivatized Cdc45 crystals were collected at the microfocus PROXIMA2 beamline of the SOLEIL synchrotron. Multiple inverse-beam segments of 60° each, separated by translation along the crystal rotation axis, were collected for each crystal at wavelength of $\lambda = 0.98012$ nm. The data were integrated with XDS[61] and three independent data sets were combined in POINTLESS and AIMLESS[62] to enhance the anomalous signal. The position of bound mercury atoms was determined using the single anomalous dispersion (SAD) method in PHENIX Autosol[63]. An interpretable electron density map was calculated to a resolution of 3.05 Å and an initial model was generated using the PHENIX AutoBuild function. The crystallographic model was extended and completed by repeated cycles of manual building in Coot[64] and crystallographic refinement with PHENIX Refine using a data set to 2.1 Å collected at Diamond Light Source, beamline i24.

The final model was refined to R-work and R-free values of 0.1679 and 0.1953 and a Molprobity score of 1.0 (ref. 65). Amino acids 137–150 of Cdc45 were not included in the final model because of missing or poor electron density and are presumed to be disordered. Amino acids S151, E152 and P153, at the N-terminus of the region from S154 to I164 that was not present in the recombinant protein used for crystallization, are visible as N-terminal fusion to residue V165. For amino acids 513–515 and 363–371, the electron density map suggested the presence of a second, alternative conformation of the polypeptide chain that was not included in the final model.

**Preparation of *Xenopus laevis* interphase egg extracts.** Female frogs were injected with 300 and 600 U of Human Chorionic Gonadotropin, respectively, 24 and 16 h before egg collection and let in a 0.1 M NaCl bath. After collection, eggs were dejellied by several washes with the Dejellying Buffer (20 mM Tris-HCl, pH 8.5, 110 mM NaCl, 5 mM DTT) and then rinsed three times with MMR buffer (5 mM HEPES-NaOH, pH 7.5, 100 mM NaCl, 0.5 mM KCl, 0.25 mM MgSO$_4$, 0.5 mM CaCl$_2$, 25 μM EDTA). Dejellied eggs were then incubated with 2 μl of 10 mM Calcium ionophore A23187 (Sigma). After 5–10 min, eggs were washed three times with MMR buffer and then rinsed twice with ice-cold S buffer (50 mM HEPES-NaOH, pH 7.5, 50 mM KCl, 2.5 mM MgCl$_2$, 250 mM sucrose) freshly supplemented with 2 mM β-mercaptoethanol and 15 μg ml$^{-1}$ Leupeptin. Activated eggs were packed by centrifugation (few seconds at 3,300$g$ in a micro-centrifuge) to get rid of excess buffer and then crushed by centrifugation for 10 min at 16,100$g$. Crude extract was separated from yolk and insoluble material, supplemented with 40 μg ml$^{-1}$ of Cytochalasin B and then ultracentrifuged for 18 min at 189,000$g$ at 4 °C with a TLA100 rotor (Beckman). The final extract, obtained by mixing the clarified protein extract and the membranes fraction was supplemented with 200 μg ml$^{-1}$ Cycloheximide (Calbiochem). Experimental procedures were approved by FIRC Institute of Molecular Oncology ethical committee IACUC and national authorities.

**Chromatin-binding assay.** In preparation for chromatin-binding studies, egg extracts were thawed on ice and supplemented with 40 mg cycloheximide (Calbiochem; CAS 66-81-9), 150 mg creatine phosphokinase (Sigma Aldrich; C3755) and 30 mM creatine phosphate (Calbiochem; CAS 19333-65-4) per ml of extract. Following careful resuspension with a cut pipette tip, the extract was split into 10 μl reactions and aliquoted into pre-chilled 0.5 ml Eppendorf tubes on ice. Demem-branated sperm DNA at a final concentration of 4,000 nuclei per μl of extract and recombinant wild-type or mutant His-Flag-tagged *Xenopus* Cdc45 protein at a final concentration of 100 ng μl$^{-1}$ were added to each sample, and mixed by gentle tapping. Control samples without DNA or protein were supplemented with corresponding volumes of EB buffer (50 mM HEPES, pH 7.5, 100 mM NaCl, 2.5 mM MgCl$_2$) or Cdc45 storage buffer (20 mM HEPES, pH 7.2, 150 mM NaCl, 5% (w/v) glycerol), respectively.

Reactions were incubated at 23 °C and DNA replication was stopped by diluting the samples 30-fold with ice-cold EB-N buffer (50 mM HEPES, pH 7.5,

100 mM NaCl, 2.5 mM MgCl$_2$, 0.25% (v/v) NP40) supplemented with 3 mM β-mercaptoethanol (BME). The diluted extract was carefully layered onto an equal volume of EB-N buffer plus 3 mM BME in the presence of 30% (w/v) sucrose. Chromatin and chromatin-bound proteins were subsequently spun through the sucrose cushion for 5 min at 8,300$g$ and 5 °C. Both the layer above the sucrose cushion and most of the cushion itself were carefully removed with a vacuum aspirator, leaving ~50 μl of cushion solution behind. To remove the remaining sucrose, the undefined chromatin pellet was washed by adding 300 μl of ice-cold EB plus 3 mM BME buffer followed by centrifugation at 9,300$g$ for 5 min at 4 °C using a fixed-angle rotor.

The supernatant was removed completely using a thin needle and syringe and 15 μl of 1 × SDS chromatin loading dye was added to each sample, for western blot analysis. The following primary antibodies were used: anti-p180 (Rabbit (Rb)_ab31777, Abcam, 1:1,000), anti-Mcm7 and anti-Orc1 (Mouse (Ms)_sc-9966 and Ms_sc-53391, Santa Cruz Biotechnology, Inc., 1:1,000 and 1:2,000, respectively), anti- H2B (Rb_07-371, Merck Millipore, 1:2,000). Primary antibodies against endogenous Cdc45, recombinant Cdc45 and Psf3 were as described inref. 59, and were used at working dilutions of 1:2,000, 1:500 and 1:1,000, respectively. The uncropped version of the western blot analysis in Fig. 8b is shown in the Supplementary Information.

**DNA replication assay.** Interphase extracts were supplemented with 15 mM creatine phosphate and 150 μg ml$^{-1}$ creatine phosphokinase as energy regenerator system. For each replication assay, extracts were supplemented with 6,000 demembranated *Xenopus* sperm nuclei per μl, 0.1 μCi μl$^{-1}$ α$^{32}$P-dCTP and 100 ng μl$^{-1}$ of recombinant wild-type or mutant recombinant Cdc45 protein. DNA replication reactions were carried out at 23 °C. Each 20 μl time point was stopped by adding 160 μl of Stop-C buffer (0.5% (w/v) SDS, 5 mM EGTA, 20 mM Tris-HCl, pH 7.5) supplemented with 0.2 mg ml$^{-1}$ Proteinase K. Percent of replicated input DNA was evaluated as published[66].

**DNA substrates for EMSA.** Single-stranded, 5′FAM-labelled DNA substrates used for electrophoretic mobility shift assays were designed and analysed for unwanted secondary structure using NUPACK[67]. Double-stranded DNA substrated were generated by annealing partially or fully complementary oligonucleotides. DNA substrates used in Cdc45-DNA-binding studies are: F_80 (ssDNA 80mer), F_80 and U_80 (dsDNA 80mer), U_40 and F_20 (5′ 20mer overhang), F_40 and U_20 (3′ 20mer overhang), F_45Y and U_45Y (25mer poly-dT y-fork), F_70Y and U_70Y (50mer poly-dT y-fork), F_80B and U_80B (40mer poly-dT bubble) and F_20H (20mer poly-dT hairpin).

| | |
|---|---|
| F_80 | 5′FAM-CACACCCAACAATCACAAACACAACTCCACCCAAACACAAACACATCCCACCACAACACTCCAACTCCCAACAACAACCC-3′ |
| U_80 | 5′-GGGTTGTTGTTGGGAGTTGGAGTGTTGTGGTGGGATGTGTTTGTGTTTGGGTGGAGTTGTGTTTGTGATTGTTGGGTGTG-3′ |
| U_40 | 5′-TTTTTTTTTTTTTTTTTTTTTCCCATCTAACTCCACTCCAC-3′ |
| F_20 | 5′FAM-GTGGAGTGGAGTTAGATGGG-3′ |
| F_40 | 5′FAM-CCCATCTAACTCCACTCCACTTTTTTTTTTTTTTTTTTTTT-3′ |
| U_20 | 5′-GTGGAGTGGAGTTAGATGGG-3′ |
| F_45Y | 5′FAM-TTTTTTTTTTTTTTTTTTTTTTTTTCCCATCTAACTCCACTCCAC-3′ |
| U_45Y | 5′-GTGGAGTGGAGTTAGATGGGTTTTTTTTTTTTTTTTTTTTTTTTT-3′ |
| F_70Y | 5′FAM-TTTTTTTTTTTTTTTTTTTTTTTTTTTTTTTTTTTTTTTTTTTTTTTTTCCCATCTAACTCCACTCCAC-3′ |
| U_70Y | 5′-GTGGAGTGGAGTTAGATGGGTTTTTTTTTTTTTTTTTTTTTTTTTTTTTTTTTTTTTTTTTTTTTTTTTTT-3′ |
| F_80B | 5′FAM-CCATCATTCCAACTCCACTCTTTTTTTTTTTTTTTTTTTTTTTTTTTTTTTTTTTTTTTCCACTTCACCATTCACT-3′ |
| U_80B | 5′-AGTGAATGGTGAAGTGGAAATTTTTTTTTTTTTTTTTTTTTTTTTTTTTTTTTTTTTTTTTTGAGTGGAGTGGAATGATGG-3′ |
| F_20H | 5′FAM-CGTAGCGGCGTCCGACGACGCCCTACCTTAAACCCACATCCGTCGTCGGACGCCGCTACG-3′ |

**Electrophoretic mobility shift assay.** The affinity of human Cdc45 for each DNA construct was tested using 1, 5, 10, 25, 50, 75 or 100 μM of TEV-cleaved human Cdc45 in 20 mM HEPES, pH 7.3, 100 mM NaCl, 5% (w/v) glycerol and 1 mM DTT. For each protein concentration, the reaction volume was adjusted to 8 μl using the same buffer and 2 μl of DNA in 1 × TE buffer was added to each reaction at a final concentration of 1 μM. Samples were incubated at room temperature for 20 min together with a control sample without protein, transferred on ice and 1 μl of 10 × EMSA loading dye (250 mM Tris, pH 7.5, 40% (w/v) glycerol, 0.25% (w/v) Bromophenol Blue) was

added. To analyse the amount of bound DNA, samples were separated by gel electrophoresis on a 1.2% (w/v) agarose gel in 0.5 × Tris-Borate buffer at 4 °C.

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

## Acknowledgements

We thank Ben Luisi for help with X-ray data collection, Alessandro Costa for sharing the cryoEM data of the CMG complex before publication and Joseph Maman for help with the analysis of Cdc45–DNA interactions. This work was supported by a Wellcome Trust Senior Investigator award to L.P. (104641/Z/14/Z) and a Cambridge Gates PhD scholarship to A.C.S. V.C. is funded by the Associazione Italiana per Ricerca sul Cancro, the European Research Council consolidator grant (614541), the Association for International Cancer Researc, the Giovanni-Armenise award to V.C., the Epigen Progetto Bandiera and the Fondazione Telethon.

## Author contributions

A.C.S. solved the crystal structure of human Cdc45, prepared the recombinant *Xenopus* Cdc45 proteins, performed the Cdc45-DNA-binding experiments and determined the optimal concentration of recombinant Cdc45 to use for the functional assays in *Xenopus* egg extracts (Supplementary Fig. 4); V.S. carried out the functional analysis of the Cdc45 mutants by performing the DNA synthesis assays and the chromatin-binding assays (Fig. 8 and Supplementary Fig. 5); A.C.S. and L.P. conceived the project; A.C.S., V.S., V.C. and L.P. designed the experiments; LP wrote the paper with input from all authors.

## Additional information

**Accession codes:** Coordinates and structure factors have been deposited in the Protein Data Bank under accession code 5DGO.

**Competing financial interests:** The authors declare no competing financial interests.

**How to cite this article**: Simon, A. C. *et al.* Structure of human Cdc45 and implications for CMG helicase function. *Nat. Commun.* 7:11638 doi: 10.1038/ncomms11638 (2016).

