## [Peer Review File · Nature Communications]

Reviewer #1 (Remarks to the Author):

Simon et al describe analysis of the X-ray crystallographic-derived structure of human Cdc45. The associated structure was then docked into a 7.4 EM map (published in Nat Comm, 2016, the Costa research group) of the Drosophila CMG (Cdc45, MCM2-7, GINS) helicase, which operates during eukaryotic DNA replication to unwind DNA ahead of the advancing replication fork. Docking of the X-ray-derived Cdc45 structure into the CMG EM volume is then used to extract insights on the interaction between Cdc45 and other proteins of the CMG helicase complex. These insights, along with others obtained about Cdc45 itself are test experimentally. Biochemical and structural analysis of the factors associated with eukaryotic DNA replication represent an important scientific area of intense interest.

A. The major findings of this work are:

- 1) X-ray crystal structure of human Cdc45. An analysis to 2.0 Å is presented. Crystallization and structure determination are first rate.

- 2) Structure of Cdc45. The authors note similarity of the Cdc45 structure to the bacterial RecJ exonuclease. This similarity had been anticipated by the bioinformatic analyses of several groups. One of these groups (S. Onesti) described an elongated conformation for Cdc45 that suggests the potential for interesting differences with the compact/disk-like structure reported here.

- 3) Docking of the X-ray structure of Cdc45 onto the 7.4Å EM map of the CMG-helicase. The human Cdc45 crystal structure was docked into an EM volume of the Drosophila CMG helicase (as described by 2016 work by the Costa group) near the gap between MCM subunits 2 and 5 of the CMG helicase; this is done using a straightforward approach. This represents an advance in that prior EM analyses had been able to derive atomic analyses of all the components of the CMG helicase save for Cdc45, because no crystal structure had been available, until the present work. It should be noted that not only was the crystal structure reported here used to enhance the atomic model of the Drosophila CMG helicase, but it was also employed to advance the analysis of the 3.7 - 4.8 Å analysis of the *S. cerevisiae* CMG helicase by Li and O'Donnell (NSMB, 2016). The authors build homology models of human MCM2 and MCM5 to obtain further insights into the interaction between Cdc45 and MCM2-7. The authors also analyze the interface between Cdc45 and the GINS ensemble. Inspection of the Cdc45 interfaces with MCM subunits and the GINS complex leads identification of several amino acids on Cdc45 as targets for mutation and biochemical analysis. The interface mutants showed strong inhibition of DNA synthesis in reactions done in *Xenopus* egg

extracts. Additional chromatin loading assays directly probe the biochemical defects of the positions of interest. Mutation of two other regions of Cdc45 (not at putative interfaces) also implies an importance for these segments in biochemical function.

4) The authors speculate on the implications of their findings for CMG assembly and function.

B. This work is original, and confirms a great deal of experimental results from other groups.

C. The data presented to support this study are of high quality; they appropriately support the conclusions that are drawn. The presentation is appropriate.

D. Use of statistics/uncertainties. In this regard, no quantitative measure that captures how well the Cdc45 structure fits into the CMG EM map is provided. However, the reader will be more than persuaded of correctness of the fit by the visual match between atomic model and EM map (Figure 4). The results provided by the mutational analysis (tested via biological and biochemical assays) serves to further reinforce the validity of the fit.

E. Conclusions. See A above

F. Suggested improvements. None.

G. References. These are appropriate.

H. Clarity and context. Appropriate.

Reviewer #2 (Remarks to the Author):

The revised manuscript from Simon et al. addresses the majority of points raised on the previous review. Moreover, the addition of a new functional assay lends strong support for the interfaces described for Cdc45 binding to MCM and GINS. While these revisions have significantly improved the manuscript, there are a few issues that still need to be resolved prior to publication.

All requested responses/revisions concern the data presented in the section, "Structure-function analysis of Cdc45":

1) The authors make several statements when discussing their results that are not wholly substantiated by the data. These include:

a. "All interface mutants showed a recruitment delay and reduced levels of Pol a and GINS" - no recruitment delay is observed for the GINS2 or MCM3 mutant in Supplemental Figure 5. Furthermore, GINS2 appears to have unaltered levels of Pol a recruitment in Supplemental Figure 5 (relative to WT), but reduced levels in the main text Fig. 8B. Were these data from separate experiments? In the absence of replicates (perhaps these have been done), band quantitation, and the relevant statistical analysis, a simple description of the results may be all that is needed to support the structural modeling: the reviewer agrees with the authors that the data in Fig. 8B clearly demonstrate that all MCM mutants show defective recruitment, and this affects Psf3 loading as expected from stepwise recruitment of Cdc45 followed by GINS. Likewise, all GINS mutations show relatively normal recruitment, but reduced levels of GINS.

b. "The remaining mutants showed normal kinetics and recruitment levels for Pol a and GINS" - The first part of the statement appears accurate: the kinetics do not, in fact, appear altered. However, the Pol a recruitment levels are most definitely altered. If MCM3 is considered to have reduced levels of Pol a recruitment in the main text figure, then so should the Basic mutant. Also, recruitment of Pol a with Helix6 is clearly increased relative to WT protein. Comment?

2) The aforementioned questions brings up an another important point. The enhanced levels of replication observed for WT Cdc45 are attributed to enhanced Pol a recruitment, consistent with the chromatin loading assay results. However, Helix6 also shows enhanced Pol a recruitment but reduced replication, and R407A has normal Pol a recruitment but reduced replication. This is an interesting feature of the data - are there any unique mechanistic insights that can be gleaned from this?

3) If the MCM mutants are defective in MCM binding (which the chromatin loading assay supports), how can they compete with endogenous Cdc45 to reduce replication below that observed for buffer? This is expected for the GINS mutants, which presumably compete with endogenous Cdc45 and bind MCM, but then poison the intermediate by preventing recruitment of GINS. An explanation of the observed reduction in replication should be provided.

Reviewer #3 (Remarks to the Author):

The revised manuscript from Pelligrini and colleagues describes a high-resolution structure of the DNA replication protein Cdc45. The availability of a higher resolution EM structure to do the docking in makes this previously problematic aspect of the paper much more clear. The authors have also improved the analysis of the mutants that they constructed in Cdc45 by including ChIP assays that directly address Cdc45 and GINS association with the DNA. This makes the conclusions about the

impact of the mutations on CMG formation much better founded. This version of the manuscript will be of significant interest to the DNA replication and cell cycle fields.

Specific point.

1. The figures in the paper showing the structure are much better annotated/connected to the text.

1. The authors should discuss the reduced Mcm7 association seen for the majority of the mutants (all except MCM1 and Helix6). Presumably this is due to a loss of Mcm2-7 association in the absence of CMG formation but this should be stated. This would suggest that despite the stability of Mcm2-7 on DNA during G1, in S phase there is a requirement for Cdc45/GINS for Mcm2-7 to be retained on chromatin.

Reviewer #1

Nothing to address.

Reviewer #2

1) *The authors make several statements when discussing their results that are not wholly substantiated by the data. These include:*

a. "All interface mutants showed a recruitment delay and reduced levels of Pol α and GINS" - no recruitment delay is observed for the GINS2 or MCM3 mutant in Supplemental Figure 5. Furthermore, GINS2 appears to have unaltered levels of Pol α recruitment in Supplemental Figure 5 (relative to WT), but reduced levels in the main text Fig. 8B. Were these data from separate experiments?

We would like to emphasise that the data in Fig. 8 and Suppl. Fig. 5 are the result of separate experiments, showing separate western blots with different exposure times, and therefore should not be directly compared. Fig. 8 is representative of experiments repeated at least three times and should be used for quantitative evaluation of protein binding to chromatin, whereas Suppl. Fig. 5 can only be used for kinetic evaluations of mutant behaviour.

Thus, analysis of the kinetic behaviour of the interface mutants in Suppl. Fig 5 shows various degrees of delay in Pol α recruitment compared to endogenous and wild-type Cdc45. The delay is most conspicuous for GINS1 and MCM1-2 and less pronounced for GINS2 and MCM3. Furthermore, Fig. 8 shows a clear reduction in Pol α recruitment to chromatin for the GINS2 mutant.

We have now rewritten the relevant section of the Results, concerning the replication assays in *Xenopus* egg extracts ('Structure - function analysis of Cdc45' section of the Results, page 17 of the revised manuscript), to improve and clarify the description of these results. We have also split over two paragraphs the description of the Results pertaining to the chromatin binding (Fig. 8) and kinetic analysis (Supplementary Fig. 5), to highlight the fact that these are separate experiments.

In the absence of replicates (perhaps these have been done), band quantitation, and the relevant statistical analysis, a simple description of the results may be all that is needed to support the structural modeling; the reviewer agrees with the authors that the data in Fig. 8B clearly demonstrate that all MCM mutants show defective recruitment, and this affects Psf3 loading as expected from stepwise recruitment of Cdc45 followed by GINS. Likewise, all GINS mutations show relatively normal recruitment, but reduced levels of GINS.

We have rewritten our description of the results in the revised manuscript, as explained above.

b. "The remaining mutants showed normal kinetics and recruitment levels for Pol α and GINS" - The first part of the statement appears accurate: the kinetics do not, in fact, appear altered. However, the Pol α recruitment levels are most definitely altered. If MCM3 is considered to have reduced levels of Pol α recruitment in the main text figure, then so should the Basic mutant. Also, recruitment of Pol α with Helix6 is clearly increased relative to WT protein. Comment?

We agree with the reviewer that the effect of the Cdc45 mutants Helix, Basic and R407 on Pol α recruitment is not entirely uniform, as we had originally stated, and we have clarified this in the text ('Structure - function analysis of Cdc45' section of the Results, page 17 of the revised manuscript).

2) The aforementioned questions brings up an another important point. The enhanced levels of replication observed for WT Cdc45 are attributed to enhanced Pol α recruitment, consistent with the chromatin loading assay results. However, Helix6 also shows enhanced Pol α recruitment but reduced replication, and R407A has normal Pol α recruitment but reduced replication. This is an interesting feature of the data - are there any unique mechanistic insights that can be gleaned from this?

We agree with the reviewer that the different mutant behaviour relative to Pol α recruitment and DNA synthesis might have an interesting mechanistic basis. At this stage, it is unclear what might be the reasons for these differences. We speculate that the effect might be due at least in part to their different ability to replace endogenous Cdc45; as we comment in the text, the Basic mutant is particularly deficient in this, pointing perhaps to its defective engagement with the machinery responsible for loading Cdc45 onto chromatin. In addition, it should be noticed that effects on levels of DNA synthesis will depend also on other replication factors, such as DNA polymerases, whose binding to chromatin has not been investigated for lack of specific antibodies.

We have added a brief comment in the Discussion (page 20 of the revised manuscript), to explain how understanding the observed functional effects of Cdc45 mutants Helix6, Basic and R407A will require further studies of Cdc45's role in CMG function and mechanism of assembly.

3) If the MCM mutants are defective in MCM binding (which the chromatin loading assay supports), how can they compete with endogenous Cdc45 to reduce replication below that observed for buffer? This is expected for the GINS mutants, which presumably compete with endogenous Cdc45 and bind MCM, but then poison the intermediate by preventing recruitment of GINS. An explanation of the observed reduction in replication should be provided.

We agree with the reviewer that the mechanism by which the MCM and GINS mutants affect replication is likely to be different. We share her/his view that the GINS mutants might act by blocking recruitment of GINS and preventing CMG assembly. We surmise that the MCM mutants also interfere with CMG assembly, but by a different mechanism, resulting from their normal transport to chromatin, which is independent of binding to the MCM proteins, coupled to their inability to associate stably with the MCM ring. By acting as competitive inhibitors of endogenous Cdc45, the MCM mutants would then prevent completion of CMG assembly.

As suggested by the reviewer, we have included in the Discussion our interpretation for the reduction in replication levels caused the MCM mutants (page 19 of the revised manuscript).

Reviewer #3

1. The authors should discuss the reduced Mcm7 association seen for the majority of the mutants (all except MCM1 and Helix6). Presumably this is due to a loss of Mcm2-7 association in the absence of CMG formation but this should be stated. This would suggest that despite the stability of Mcm2-7 on DNA during G1, in S phase there is a requirement for Cdc45/GINS for Mcm2-7 to be retained on chromatin.

There is no clear separation between G1 and S-phase in *Xenopus* egg extract (basically there is no G1), and CMG is continuously assembled at new replication origins throughout the duration of S-Phase. As the reviewer suggests, it is possible that the

reduced levels of CMG on chromatin observed for some of the Cdc45 mutants might be caused by an increased rate of CMG unloading.